# First evidence of active transpressive surface faulting at the front of the eastern Southern Alps, northeastern Italy. Insight on the 1511 earthquake seismotectonics

Emanuela Falcucci[1], Maria Eliana Poli[2], Fabrizio Galadini[1], Giancarlo Scardia[3], Giovanni Paiero[2] Adriano Zanferrari[2]

[1]Istituto Nazionale di Geofisica e Vulcanologia, Roma, Via di Vigna Murata 605, 00143, Italy
[2] University of Udine – Dept. of Agricultural, Food, Environmental and Animal Sciences, Udine, Italy.
[3] Universidade Estadual Paulista (UNESP), Instituto de Geociências e Ciências Exatas, Rio Claro – SP, Brazil.

*Correspondence to*: Emanuela Falcucci (emanuela.falcucci@ingv.it)

**Abstract.** We investigated the eastern corner of northeastern Italy, where a system of NW-SE trending dextral strike-slip faults of western Slovenia intersects the south-verging fold and
thrust belt of the eastern Southern Alps. The area suffered the largest earthquakes of the region, among which are the 1511 (Mw 6.3) event and the two major shocks of the 1976 seismic sequence, with Mw=6.4 and 6.1. The Colle Villano thrust and the Borgo Faris-Cividale strike-slip fault have been here first analyzed by interpreting industrial seismic lines and then by performing morpho-tectonic and paleoseismological analyses. These different datasets indicate that the two structures define an active, coherent transpressive fault system that activated twice in the past two millennia, with the last event occurring around the 15th-17th century. The chronological information, and the location of the investigated fault system suggest its activation during the 1511 earthquake.

**Keywords.** active transpressive tectonics, surface faulting, paleoseismology, 1511 earthquake, eastern Southern Alps.

## 1 Introduction

The Late Miocene-Quaternary counterclock-wise rotation and contemporaneous northward drift of the Adria microplate indenter resulted in the development of the eastern Southern Alps, which are connected to the Dinarides towards the east. The Adria microplate kinematics determined diffuse dextral strike-slip deformation in Slovenia and prevailing thrusting at the eastern Southern Alps, in northeastern Italy (Zanferrari et al., 2013). Seismicity reflects such a kinematic transition, being characterized by both earthquakes caused by dextral strike-slip
and reverse ruptures (Kastelic et al., 2008). This issue is by all means relevant considering that this region has been the focus of some of the strongest historical earthquakes of continental Europe, among which are the 1348 (Mw 6.6) and the 1511 (Mw 6.3) events, as well as the two major shocks of the 1976 seismic sequence (Mw 6.4 and 6.1). In particular, despite the large number of studies (e.g., Ambraseys, 1976; Ribaric, 1979; Bavec et al., 2013), the epicentre, the causative fault(s) and the kinematics of the 1511 earthquake are still a matter of debate. Here we describe the results of a multi-disciplinary study performed in the 1511 earthquake area, based on geological-geomorphological surveys, industrial seismic lines interpretation, paleoseismological trenching and the drilling of a 20 m-deep core. Specifically, we focus on the Borgo Faris-Cividale Fault (henceforth BFCF), a dextral strike-slip
structure that experienced a complex kinematic history (e.g., Zanferrari et al., 2008; Zanferrari et al., 2013), and the Colle Villano Thrust (henceforth CVT), that shows geomorphic hints of recent activity (Galadini et al., 2005). We aim to understand the relationship between these very close structures and their role in the regional structural-kinematics framework, and to acquire new clues on the 1511 earthquake seismotectonics.

## 2 Tectonic setting and seismic activity

Since the Middle Miocene, the SSE-verging and WSW-ENE-trending fronts of the eastern Southern Alps in the Friuli region (Fig. 1a) (e.g., Castellarin et al., 2006, and reference therein)
cut and re-folded the external Paleogene Dinarides compressive structures (e.g., Doglioni and Bosellini, 1987; Zanferrari et al., 2013). At the Miocene-Pliocene transition, the counterclockwise rotation of the Adria microplate produced dextral strike-slip deformation in Slovenia (e.g. Marton et al., 2003; Vrabec and Fodor, 2006). Recent activity of dextral strike-slip fault systems is documented by large right-lateral offset of geological and geomorphological features (e.g., Mlakar, 1969; Cunningham et al., 2006, 2007; Moulin et al., 2014, 2016) and by the formation of pull-apart basins (Vrabec, 1994; Kastelic et al., 2008). Seismicity also reflects dextral strike-slip deformation, with major earthquakes having transcurrent focal plane solutions (e.g., Poljak et al., 2010), such as the April 12, 1998 (Mw 5.66) (Bajc et al., 2001; Zupancič et al., 2001) and July 12, 2004 (Mw 5.1) earthquakes, related to the
Ravne strike-slip fault (Kastelic et al., 2008). Quaternary activity of the eastern Southern Alps front is documented by field evidence and seismic lines interpretation (e.g., Zanferrari et al.,1982; Galadini et al., 2005; Zanferrari et al., 2008a-b; Poli et al., 2008; Poli et al., 2009; Zanferrari et al., 2013; Monegato and Poli, 2015), which defined ongoing growth of anticlines on blind active thrusts. The two aforementioned main shocks of the 1976 seismic sequence, in central Friuli, show reverse fault plane solutions, along low angle WNW-ESE to E-W striking and N-dipping reverse faults (Fig. 1a) (Slejko et al., 1999; Pondrelli et al., 2001; Poli et al., 2002), recently confirmed by geodetic data analysis and inversion (Cheloni et al., 2012). Interseismic geodetic data show about 2 mm/yr northwards movement of Adria relative to Eurasia (e.g., D'Agostino et al., 2008; Devoti et al., 2011; Carafa and Bird, 2016) (Fig.
1a). This is absorbed by WSW-ENE trending, SSE-verging thrust front of the eastern Southern Alps, and by NW-SE trending, right-lateral strike-slip fault systems in western Slovenia. The major historical earthquake of the study area struck on March 1511 (maximum intensity = IX° MCS scale). In spite of many studies, many issues still remain to be solved about this event. Ambraseys (1976) suggested M~6.4 and epicentre located northwest of Tolmin, at the Italy-Slovenia border. Ribaric (1979) suggested that the event have been actually made of two shocks, one occurred at 15h CET in the Idrija zone, in Slovenia, with possible magnitude 6.9, and a second at 20h CET east of Gemona, in Friuli, with possible magnitude 7.0-7.2. Košir and Cecić (2011) questioned Ribaric's interpretation of the historical information and proposed a single main shock on March 26, at 14:40 GMT. By inverting macroseismic data,
Fitzko et al. (2005) hypothesized a possible source of the 1511 earthquake on a 50 km-long segment of the Idrija fault, in Slovenia. The authors proposed NW-ward rupture directivity, with nucleation just to the SE of the Idrija town. This hypothesis is also assumed by the Italian Database of Individual Seismogenic Sources (Basili et al., 2008). Nonetheless, as reported by Fitzko et al. (2005), their model only partly reconciles the actual intensities suffered by many villages in Italy and Slovenia. Indeed, some synthetic intensity data-points differ of up to 2 degrees from the intensities estimated by the historical sources. Moreover, a recent reappraisal of macroseismic data led to a new distribution of intensities (Camassi et al., 2011), where values are strongly decreased in Slovenia. In particular, the intensity of X assigned to Idrija, which was a key point in the Fitzko et al.'s hypothesis, has been removed. Also, Camassi et
al. (2011) proposed a new epicenter for the 1511 event in Italy, near Tarcento, and Rovida et al. (2016) defined Mw 6.3.

## 3 Structural observations and seismic line interpretation

The BFCF is a ~25 km-long, NW-SE striking dextral strike-slip fault, traceable from Nimis, to the north, to Cividale, to the south (Fig. 1a; 1b) (Moulin et al., 2016). Southwest of the BFCF occurs the CVT, a 10 km-long, WNW-ESE striking thrust. Kinematic indicators (calcite slickenlines) show a SW-ward vergence (Fig. 1b, inset). The fault front crops out at the base of small reliefs made of early Eocene turbidites (Savorgnano Marls and Arenites in Zanferrari et al., 2008a) (Fig. 1b), which have been folded and uplifted by the thrust activity. The
BFCF and CVT merge towards the SE (Fig. 1b). Interpretation of an industrial seismic line (kindly provided by ENI E&P) allowed us to define the deep geometry of the two structures (Fig. 2). The CVT cuts the Quaternary succession and seems to be connected at depth with the BFCF, representing a branching from the same major structure. Two further thrusts

(Premariacco thrust and Tarnovo thrust, i.e. PRM and TN in Fig.1b, respectively) are also interpretable in the CVT footwall, deforming the base of the Quaternary. The seismic reflection line also shows the CVT reaching the surface. Moving from this evidence, we focused paleoseismological investigation along the CVT surface trace with the aim of constraining the recent movements of the fault.

## 4 Morpho-tectonic evidence

The sector between the CVT and BFCF is characterized by a low gradient morphology, with flat sectors interposed to small NE-SW elongated gentle reliefs. The streams run from the NE to the SW, and  get sinuous entering this low gradient sector.

On the basis of morphological observations, Moulin et al. (2014) and Moulin et al. (2016) consider BFCF as an active fault, i.e. the northern portion of the Raša fault. In particular, in the study area morpho-structural evidence such as suspended Quaternary glacies, diversions and deflections along the Valle, Poiana and Meris rivers and a series of aligned gaps (Zanferrari et al., 2008; Pascolino, 2014), suggest dextral horizontal movements of strike slip fault (Fig. 3a).

Moving toward the SW (i.e. on the CVT hanging wall), because of the common water regulations, most of the rivers become rectilinear, getting sinuous again flowing toward the Friulian plain. Such a geomorphic setting suggests the formation of a low gradient sector at the CVT rear, owing to the progressive growth of the reverse tectonic structure. The presence of two back-tilted surfaces located at the boundary between the Friulian plain and the reliefs (Fig. 3a) corroborates this interpretation.

Moreover, we found remnants of an old paleo-landscape on top of the ridges located between the CVT and BFCF, represented by almost flat landsurfaces carved onto the turbidite bedrock. Interpolation of these top relict landsurfaces (Fig. 3b) indicates NE-wards dipping, that is opposite to the present drainage pattern.

In order to find further evidence of the recent activity of the CVT, we made a core boring 20 m deep just northward of the trenches site (location in Fig. 3a), above an about 5 meters raised fluvial terrace. The borehole (localized at 155 m a.s.l.) found about 3 m-thick colluvial sandy silt. Underneath, 12 m-thick grey-blue lacustrine deposits were cored. The drilling reached the bedrock (i.e. Savorgnano Marls and Arenites) at 15.40 m depth (Fig. 4).

Comparing the depth of bedrock in the trenches (unit 8) with that in the borehole it appears that the Savorgnano Marls and Arenites constitute a morpho-structural high whose progressive growth formed a local depocentre at its back (i.e. a piggy back basin), where a small lacustrine basin constituted by the grey-blue clays (Fig. 4). Within these deposits has been found a
wood that has provided an age > 45.000 BP (radiocarbon cal. age, 2σ).

## 5 Paleoseismological investigations along the CVT

We dug three trenches across a gentle surface scarp (~0.5 m high) seen at the CVT front (Figs. 3a, 5a). The excavations exposed a continental sedimentary sequence, mainly consisting of fluvial and slope deposits that we subdivided into 8 stratigraphic units (Figs. 5, 6) here described:

Unit 1: ploughed soil, made of brownish silt with sparse cm-size polygenic pebbles.
Unit 2: colluvial deposit made of yellowish/brownish sandy silt with sparse cm-sizepebbles and charcoals fragments.

Unit 3: colluvial deposit made of brownish massive sandy silt containing cm-size polygenic pebbles (mostly organised in gravel lenses), charcoal fragments and Fe-Mn concretions.

Unit 4: alluvial deposit made of clast-supported gravel with brownish silty matrix.

Unit 5: colluvial deposit made of massive yellowish-brownish sandy silt containing cm-size polygenic pebbles (mostly organised in gravel lenses), charcoal fragments and Fe-Mn concretions.

Unit 6: colluvial deposit made of yellowish and locally brownish clayey silt with sparse clasts (10 cm maximum size). The deposit underwent pedogenesis which altered the surface of the clasts and the whole sediment structure, and determined the formation of Fe-Mn concretions.

Unit 7: alluvial deposit made of polygenic gravel (cm-size pebbles) laterally grading to clayey silt with sparse pebbles. The pebbles lithology attests that the deposit has been fed by the Tagliamento River catchment.

Unit 8: bedrock represented by the Savorgnano Marls and Arenites (Ypresian, Early Eocene).

Chronologic constraints were provided by radiocarbon dating on charcoals found within the units (dating made by INNOVA SCARL laboratory) (Table 1). In this term, it must be underlined that the obtained ages all refer to charcoals, that have been included and transported by the alluvial and colluvial deposits from which we collected them. Therefore, the ages can be similar to each other or sparse. In light of this, hence, we have only considered the most recent ages achieved for each units as a *terminus post quem* for the unit deposition and, thus, for the deformation events.

The trenches show the whole stratigraphic succession warped (upward convexity) in coincidence with the surface scarp (Figs. 5b, c). The lowermost Units 7 to 4 show a slightly tighter
bending than the upper ones (Units 3 to 1). The very localized bending, the coincidence with the surface scarp, and the sedimentological interpretation rule out that this geometrical feature relates to the original depositional attitude of the layers. This is particularly evident for the fluvial Unit 7, whose attitude is expected to be sub-horizontal. Besides this evidence, each excavation showed other features (fractures and shear planes), described below, that can be associated to events of tectonic deformation (Figs. 5b, 5d, 5e, 5f, and 6 a-c).

Trench 1 (Figs. 5b and c; Fig. 6a): unit 8 (turbidite bedrock) showed pervasive cleavage with sub-vertical planes about E-W striking, indicative of localized shearing. Slope deposits of Unit 6 is unconformably overlaid by Unit 5. This suggests progressive deformation of the sequence during deposition, with the formation of angular unconformities, i.e. growth strata.
Where the sedimentary sequence displayed warping in coincidence with the surface scarp (~0.5 m high), Units 5 and 4 were also displaced by a low angle shear plane. The displacement indicates reverse kinematics, with sense of motion towards the SW (Figs. 5b, 6a). The deformation was also accommodated by a secondary reverse shear plane with opposite sense of displacement. These features were localized where the turbidite bedrock was affected by cleavage, thus demonstrating the presence of a well-developed shear zone active previously.

Trench 2: we identified high angle shear planes that offset Units 4 and 5 with an extensional kinematics, and that were sealed by Unit 3 (Figs. 5d, 6b). The geometrical characteristics of the displaced units and the coincidence with the warped portion of the succession indicate that these shear planes define tension cracking related to bending, interpreted as an extrados-
related feature (i.e. bending moment fracturing) due to a sudden warping event of the paleo-topographic surface. This event occurred after deposition of Unit 4 and before Unit 3.

Trench 3: comparably to trench 2, Units 4 to 6 are disrupted by an tensional fracture which, in turn, was sealed by Unit 3 (Fig. 6c). Moreover, in the easternmost part of the excavation, Unit 6 is brought into lateral contact with Unit 8 (turbidite bedrock) by a sub-vertical shear plane (Figs. 5e, f and 6c). This structural feature is sealed by unit 5. Furthermore, in this sector the basal contact of Unit 5 on the underlying Unit 8 gets slightly convex upward (Fig. 5e), suggesting that Unit 5 underwent slight uplift after deposition.

The described evidence allows distinguishing at least three subsequent events of deformation: the oldest event, named E3, is documented by the displacement of Unit 6 along the sub-
vertical shear plane which placed it into contact with the bedrock (seen in trench 3) and was sealed by Unit 5. E3 was thus responsible for the first surface faulting. The angular unconformity that separates Unit 5 from Unit 6 (described in trench 1) also supports the occurrence of E3, as Unit 6 has been deformed and tilted towards the SW before the deposition of Unit 5, determining an onlap geometry.

A subsequent event, named E2, is testified by primary and secondary tectonic features, i.e. the reverse fault planes (seen in trench 1), which offset the sequence up to Unit 4, and the extrados fractures (seen in trenches 2 and 3), that developed after Unit 4 deposition and before Unit 3 deposition, respectively. It is worth noting that notwithstanding extrados fractures
are secondary surface effects, their formation requires sudden warping. Otherwise, slow and progressive deformation would have been "absorbed" by a continuous deformation of the sediments. The occurrence of E2 is also suggested by the upward bending of Unit 5 overlaying the bedrock (Fig. 5e).

The latest event, named E1, is documented by the gentle warping of Units 3 to 1 (seen in all of the trenches), which matches the bending radius of the surface scarp. As units 3–1 display a lower bending than the underlying units 7–4, it testifies that the older units underwent a larger, cumulative deformation produced by E2 + E1.

The radiocarbon ages allow us to constrain E3 before the 5th millennium B.C., based on the ages obtained from charcoals collected within Unit 5, which sealed the event. As for E2 it may be constrained between the $5^{th}$ and $6^{th}$ century AD. In particular, charcoals found within Units 4 and 3 – the former displaced by E2 and the latter sealing E2 – provided a radiocarbon age ~$6^{th}$ century AD. Even if the radiocarbon age obtained from the charcoal collected in Unit 3 represents a *terminus post quem* for the unit deposition, the similarity between its age and the age obtained from the charcoal collected in Unit 4 (i.e. $6^{th}$ century AD) allows to hypothesise that E2 likely occurred around this period. Lastly, E1 took place after the $15^{th}$ century AD, based on the youngest radiocarbon age of charcoals found within Unit 2.

## 6 Discussion and concluding remarks

We performed multiple investigations on the Colle Villano Thrust (CVT) and the Borgo Faris-Cividale strike slip fault (BFCF). These structures located at the intersection between the Slovenian dextral strike-slip active shear zone and the external active thrust front of the eastern Southern Alps. Our main goal was to investigate how active tectonic deformation distributes in this region of kinematic transition and to improve the seismotectonic knowledge of the area, still incomplete in some important aspects, such as the causative fault of the largest earthquake of the study region, occurred on 1511.

Field observations coupled with the interpretation of a commercial seismic reflection line indicate that the BFCF and CVT gave rise to a major NW-SE-to-WNW-ESE striking transpressive shear zone that accommodates reverse-oblique deformation. This interpretation fits the GPS time series available for the area, which define main N-S trending shortening. Therefore, a significant horizontal shear component is inherently expected on structures obliquely oriented with respect to the N-S trending regional σ1, i.e. the axis of maximum compression. In terms of kinematic relation between the two faults, the following evidence suggest that they are the surface expression – as fault splays – of a complex fault system that accommodates transpressive tectonic deformation affecting this region: *i)* the narrow spacing (in plan view) between the two structures (towards the south, the two structure merge, as we depicted in Fig. 1); *ii)* the deep structural arrangement, achieved by the interpretation of the provided seismic lines, which suggests that the Colle Villano Thrust is a rather superficial splay that connects to the Borgo Faris-Cividale Fault and does not cut across it; and iii) the transpressive deformations we observed along the trench walls (testified by both compressive faults and deformations, and sub-vertical strike-slip shear planes), point to the Borgo Faris-Cividale Fault as major strike-slip fault, which accommodates the horizontal tectonic deformation, and the Colle Villano Thrust as a synthetic splay that accommodates the contractional component. The evidence of active deformation we found along the CVT and the available knowledge on the kinematics of the region suggest that the transpressive slip probably splits on the investigated structures, that is, mainly strike-slip along the BFCF and mainly compressive along the CVT (Fig. 7). Slip partitioning on splays of oblique structures has been observed in many cases from across the world, both as for the coseismic and long-term displacements (e.g., Wesnousky and Jones, 1994; Walker et al., 2003; King et al., 2005). In tectonic-structural perspective, our inferences match the geodetic observations made by Devoti et al. (2011) who, based on GPS time series, issued a certain amount of horizontal shear in this region. Moreover, Montone and Mariucci (2016) show that the contemporary stress map of Italy defines that this region locates at the transition between strike-slip faulting and thrust faulting, and transpressive deformation is expected.

Trench investigations across the CVT attested at least three activation events. The presence of low angle reverse faults, the displacement of some stratigraphic units along sub-vertical shear planes and the occurrence of secondary extrados fractures are indicative of sudden deformation events along the CVT, responsible for primary surface faulting. In detail, chronological data attested the penultimate event E2 likely around the $6^{th}$ century AD and the last event E1 after the $15^{th}$ century AD. E1 has been responsible for bending, that caused ~0.5 m high (minimum) surface scarp.From a seismotectonic viewpoint, the only known post-$15^{th}$ century AD earthquake of the area that has had a magnitude large enough to result in such a significant deformation is that occurred in 1511. In this perspective, basing on the regressions of Wells and Coppersmith (1994), the magnitude of the earthquake, i.e. 6.3 (Camassi et al., 2011; Rovida et al., 2016) is consistent with the activation of the 25 km long CVF-BFCF system. Therefore, the CVT-BFCF system appears as a very plausible candidate for having played a primary role in the seismogenic process of the 1511 seismic event (Fig. 8). Ultimately, this study raises significant issues on a potential major seismogenic source of a region where interseismic coupling suggests elastic strain is building up at seismogenic depths which will be released in future large earthquakes (Cheloni et al., 2014; Serpelloni et al., 2016).

**Author contribution**

Emanuela Falcucci, first and corresponding author, led the paleoseismological investigations, manuscript writing and discussed the seismotectonic interpretation; Eliana Poli and Adriano Zanferrari performed the geological, morphological and structural analysis and interpretation of the reflection seismic line; Giancarlo Scardia contributed to the stratigraphic interpretation of the trench walls; Giovanni Paiero contributed to the trenching activity; seismotectonic interpretation was discussed and shared with Fabrizio Galadini. All of the authors discussed the paleoseismological data and general aspects concerning the regional tectonic framework.

**Acknowledgments**

The authors are grateful to the Reviewers Dario Zampieri and Luke Wedmore to improve our manuscript and work with their constructive and helpful comments and suggestions. The study has been funded by the ASSESS project, Regione Friuli Venezia Giulia-Istituto Nazionale di Geofisica e Vulcanologia (INGV) agreement, Project Manager for INGV Dr. Fabrizio Galadini. The borehole was made by means CARG Project funds managed by Prof. A. Zanferrari. The authors thank Dr. Stefano Gori for the helpful discussion on active tectonics, structural, and paleoseismological issues and Dr. Andrea, Marchesini for helping us in figures preparation.

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

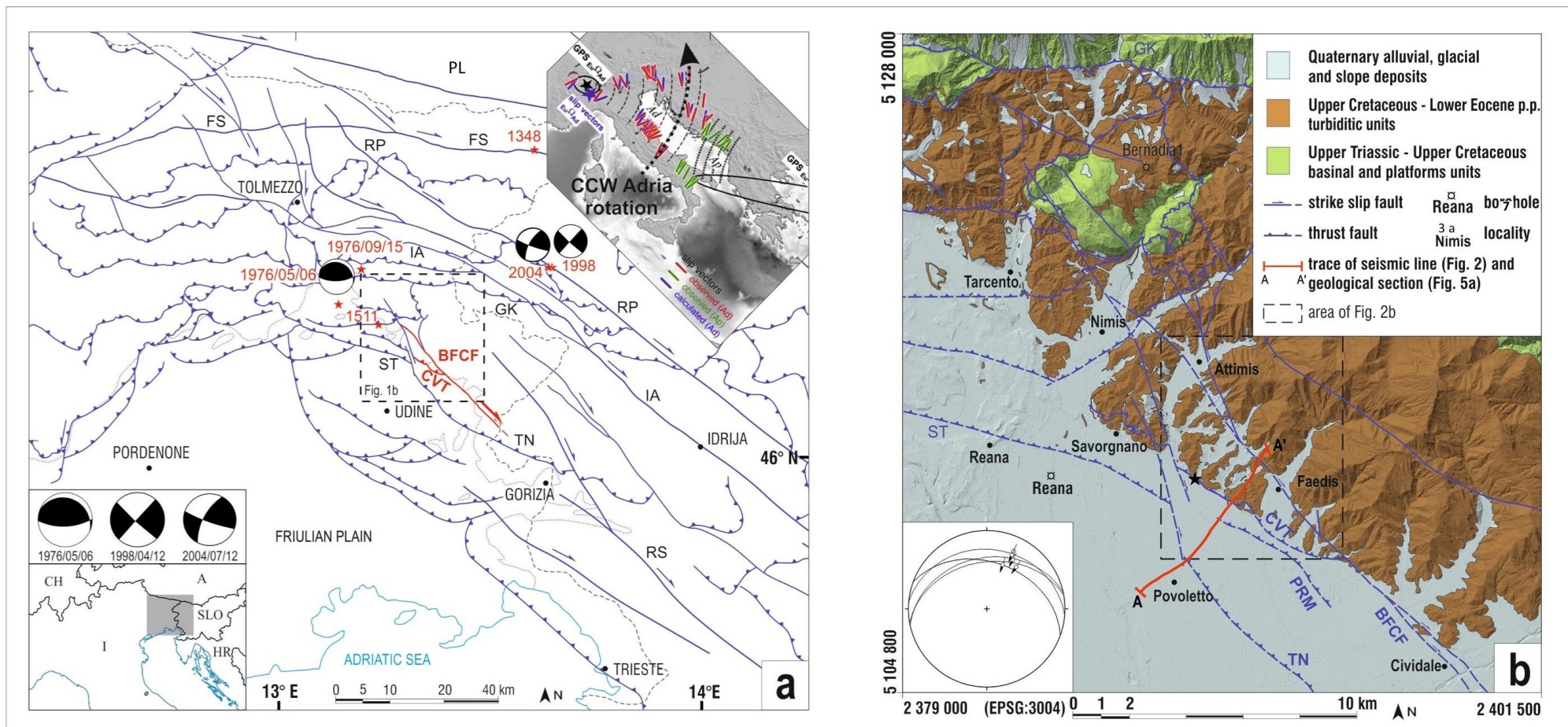

**Figure 1: a) Tectonic map of the eastern Southern Alps and western Dinarides (modified from Zanferrari et al., 2013). Adria CCW rotation (D'Agostino et al., 2008), inset; BFCF: Borgo Faris – Cividale fault; CVT: Colle Villano thrust; GK: Gemona-Kobarid thrust; IA: Idrija-Ampezzo fault; PL: Periadriatic lineament; RP: Ravne-Paularo fault; RS: Raša fault; ST: Susans-Tricesimo thrust; TN: Trnovo nappe thrust front (Placer et al., 2010). Red stars: epicentres of the strongest historical and instrumental earthquakes (Rovida et al., 2016) and the related focal plane solutions. Italian boundary, thin dashed line. Hills-plain boundary, dotted lines. b) Geological map of the study area (modified from Carulli, 2006; Zanferrari et al., 2008a; 2013). PRM: Premariacco thrust. Paleoseimological trenches site, black star. Stereographic projection (lower hemisphere) of calcite slickenlines collected on the CVT, inset.**

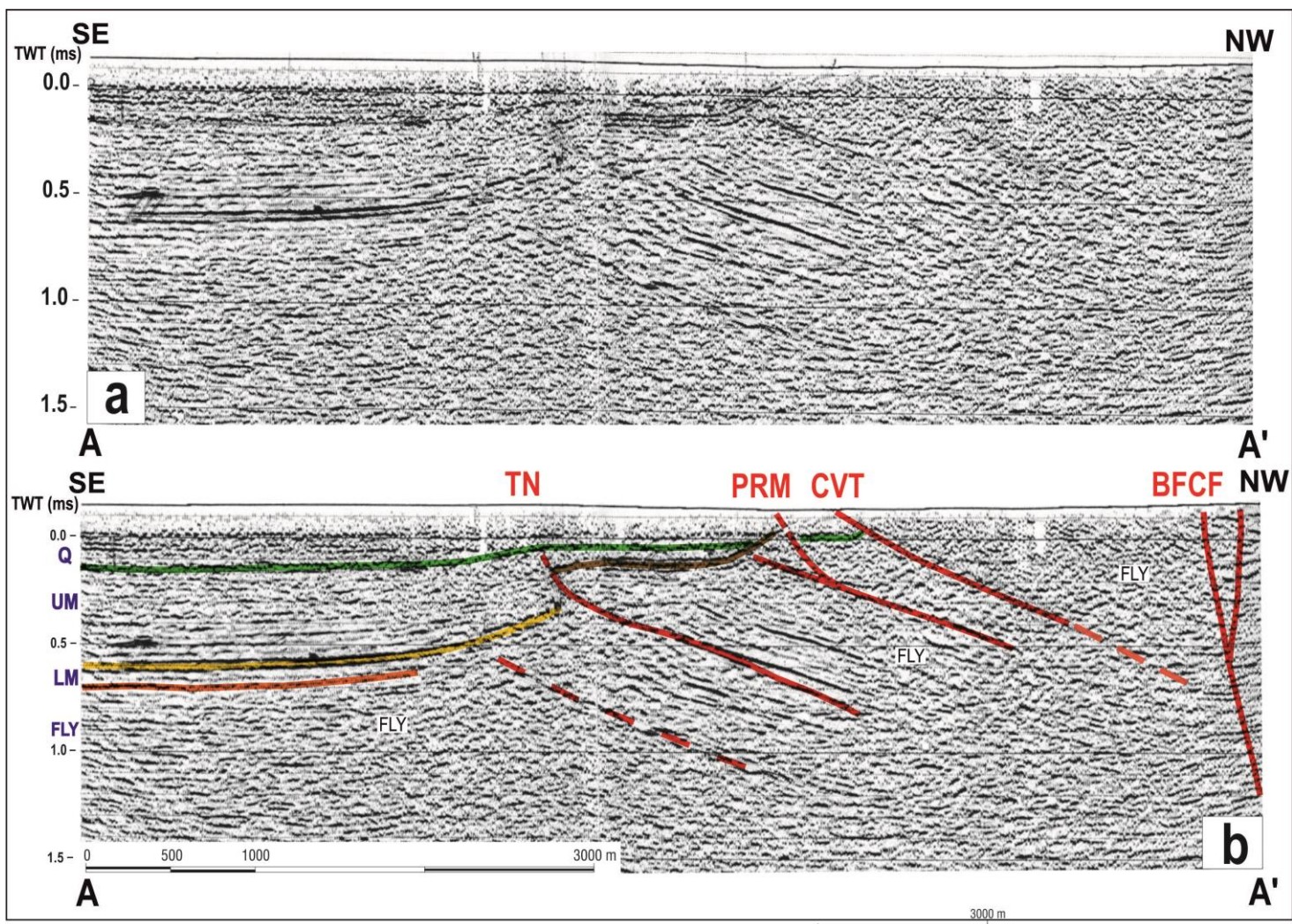

**Figure 2: a) Reflection seismic section crossing the study area; b) Interpretation (line drawing) of the reflection seismic section (A-A' in Fig. 1b). Q: Quaternary; UM: Middle-Upper Miocene Molasse; LM: Cavanella Group (Lower-Middle Miocene); FLY: Upper Cretaceous-Lower Eocene p.p. turbiditic units.**

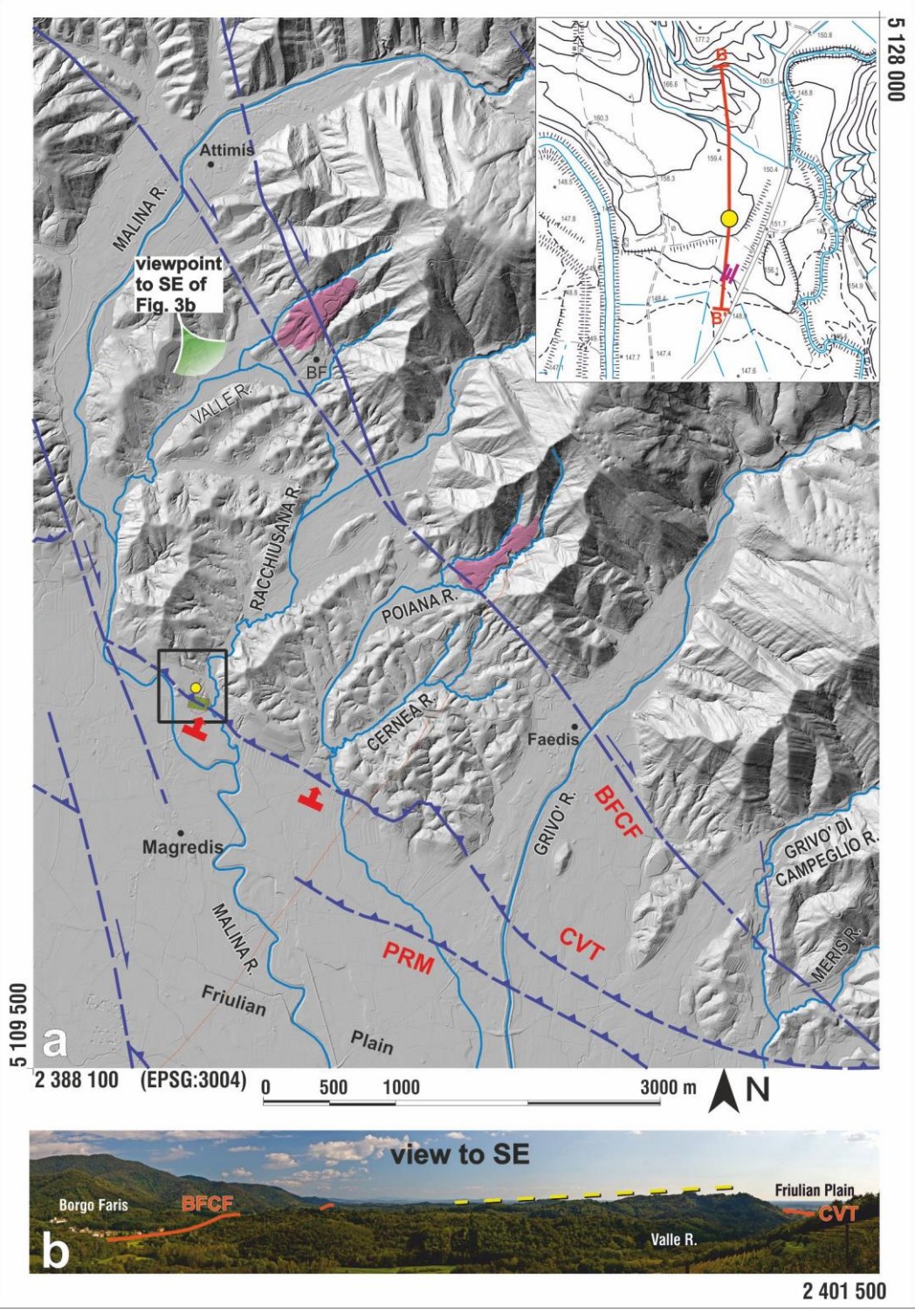

5    **Figure 3: (a) Digital Elevation Model (supplied by Friuli Venezia Giulia Region) of the study area. Faults: BFCF; CVT; PRM, Premariacco thrust; BF: Borgo Faris village. Back-tilted surfaces at the Racchiusana and Poiana valleys outlet, red arrows. In pink the two suspended Quaternary glacis cut off by the BFCF. The black square is detailed in inset: site of the core logging, yellow dot; traces of the paleoseismological trenches, violet lines, BB', geological section of fig. 4. Red line: seismic line of fig. 2. (b) The NE dipping paleolandscape carved in the turbidite bedrock (yellow dotted line) between the BFCF and the CVT. Point of view in Figure 3a (green eye).**

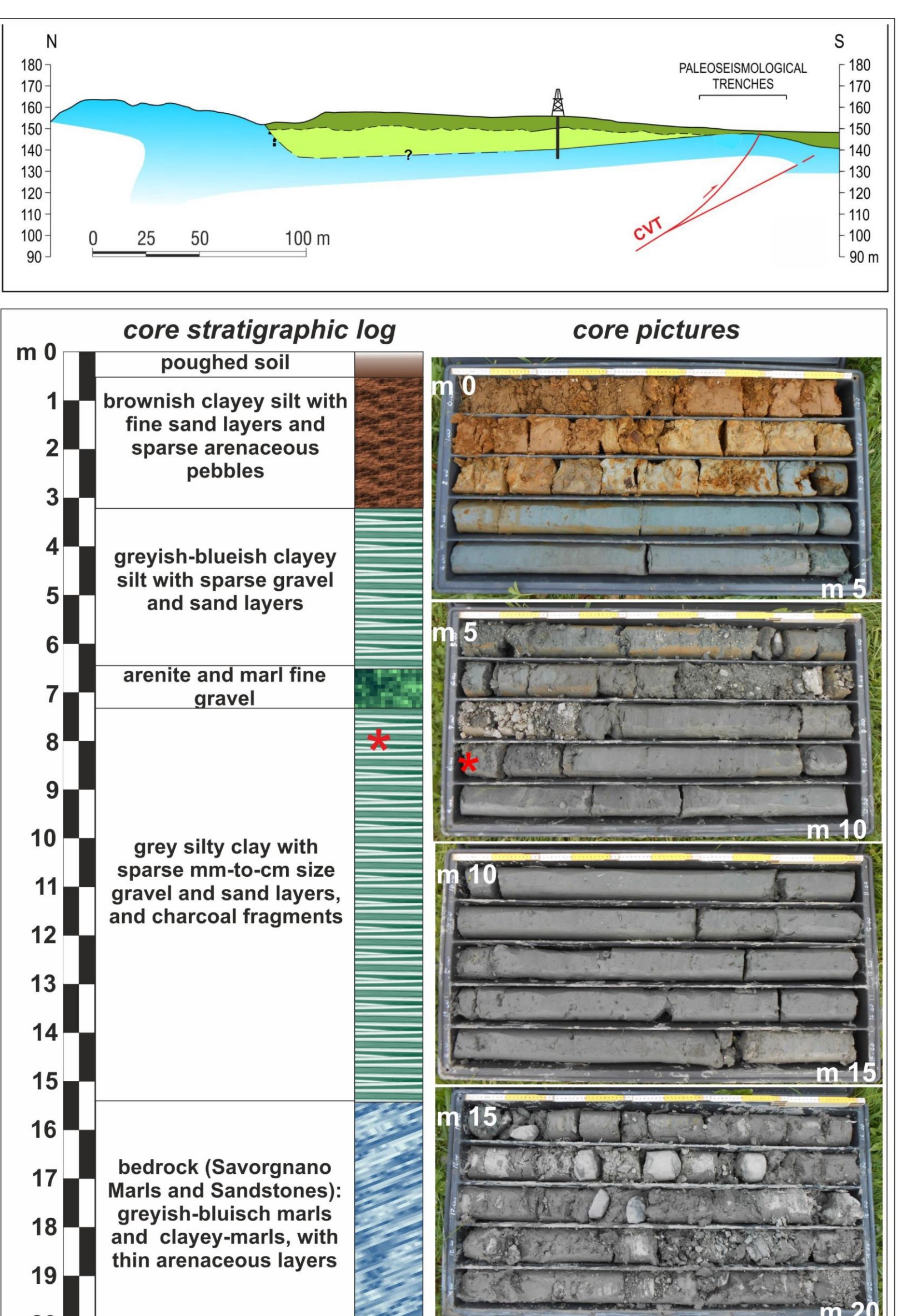

Figure 4: Geological cross-section across the core logging and the paleoseismological trenches. The light green lacustrine clay doesn't crop out in the trenches but on-laps the growing anticline built in the turbiditic bedrock (light blue). Dark green: alluvial and colluvial deposits; light green: lacustrine deposits; blue: turbidite bedrock. In the lower panel the stratigraphic log and pictures of the borehole. Red asterisk indicates the location of the sample which gave a radiocarbon age >45.000 years. Borehole location: 2389338 E, 51122357 N (EPSG: 3004).

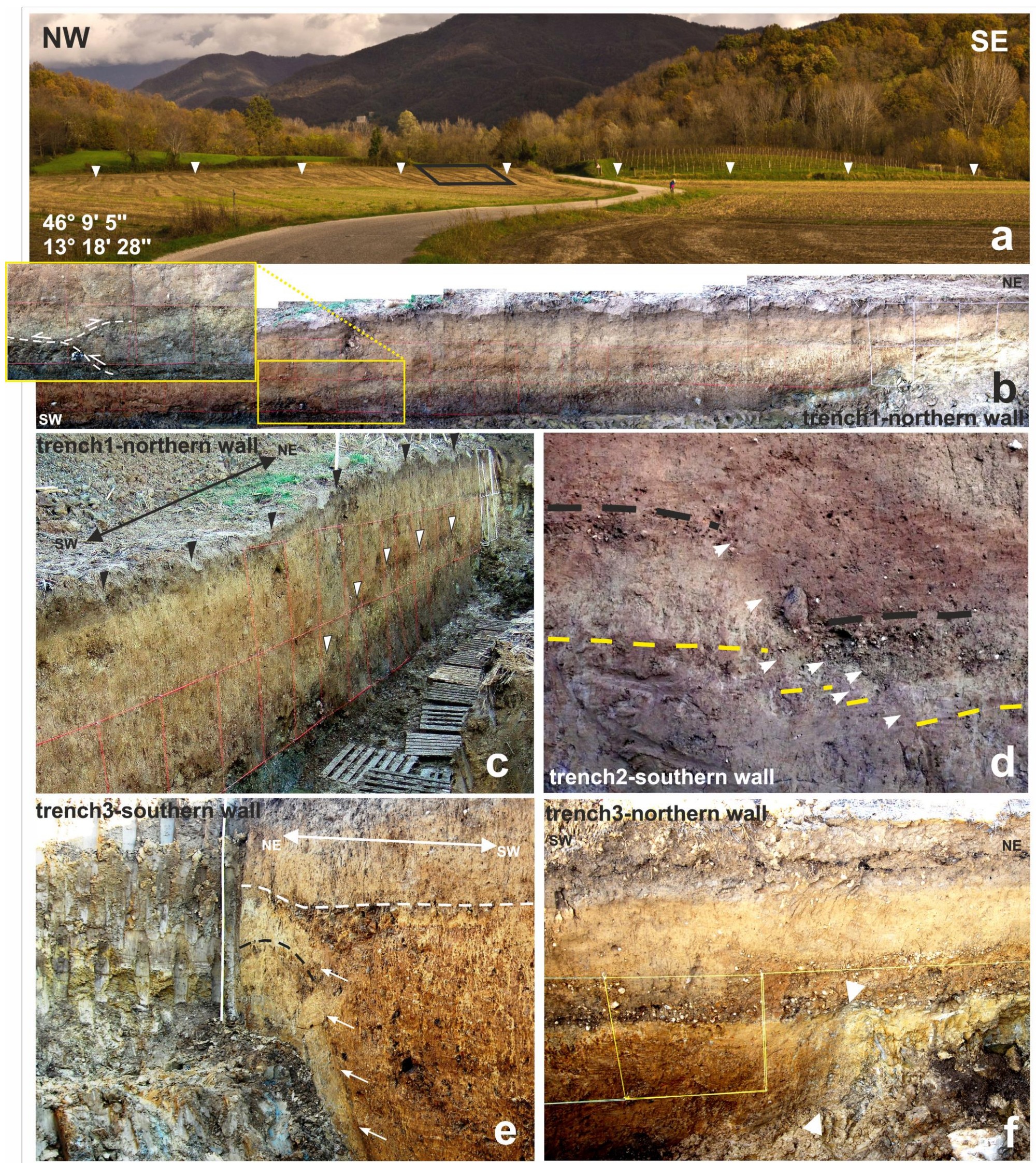

**Figure 5: a)** Racchiusana valley outlet, north of Magredis. Trenches location, black rectangle. **(b)** Trench 1, northern wall; reverse fault planes (white dashed lines in inset). **(c)** Trench 1, northern wall; bending (marked by white triangles) of the stratigraphic units in coincidence with the surface scarp (black triangles). **(d)** Trench 2, southern wall; fracture planes (indicated by white arrows) displacing the units (attitude marked by black and yellow dashed lines. **(e)** Trench 3, southern wall; shear plane (white arrows) displacing the upward warped stratigraphic units (black and white dashed lines). **(f)** Trench 3, northern wall, high angle shear plane (white arrows) placing into contact the bedrock (unit 8) with the late Quaternary units.

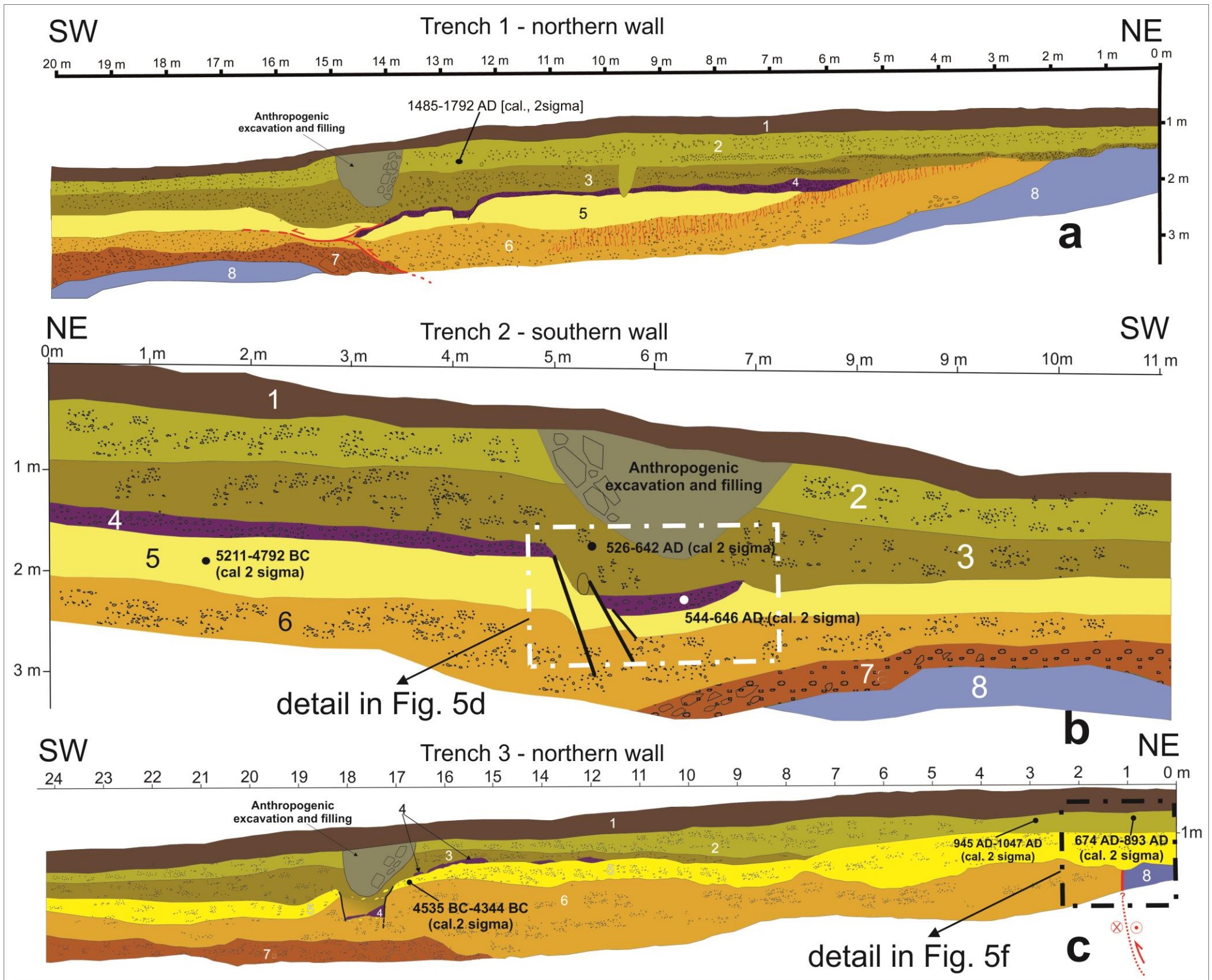

Figure 6: Trench walls, stratigraphic schemes. Units: 1, soil; 2, 3, 5 and 6, colluvial deposits; 4 and 7, fluvial deposits; 8, turbiditic bedrock.

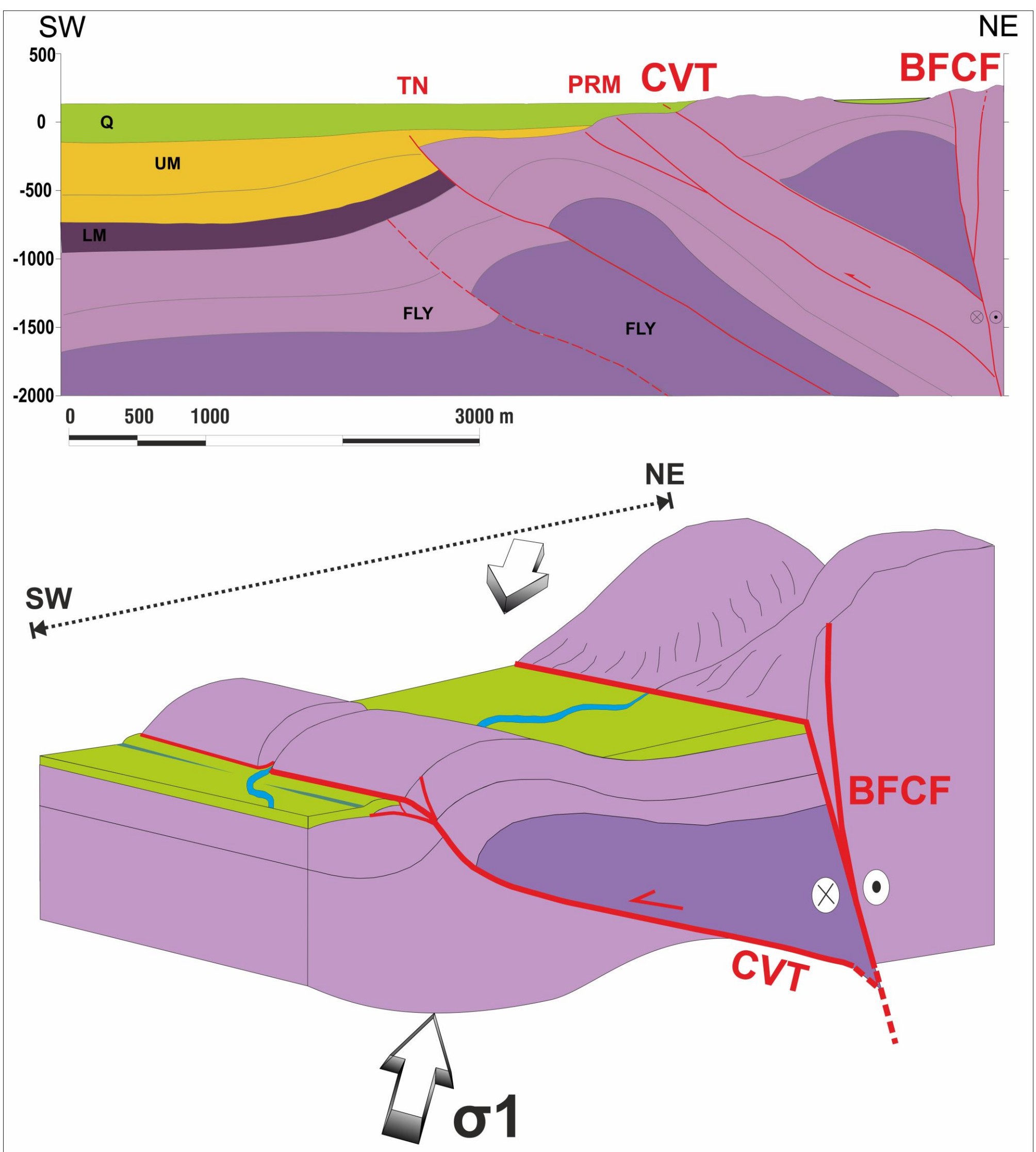

**Figure 7: Geological cross section based on the seismic line of Figure 2 (AA' in Figure 1b) and 3-D scheme (lower panel) of the BFCF-CVT system. Q: Quaternary; UM: Middle-Upper Miocene Molasse; LM: Cavanella Group (Lower-Middle Miocene); FLY: Upper Cretaceous – Lower Eocene turbiditic sequence. TN: Tarnovo Nappe front (according to Placer et al., 2010)**

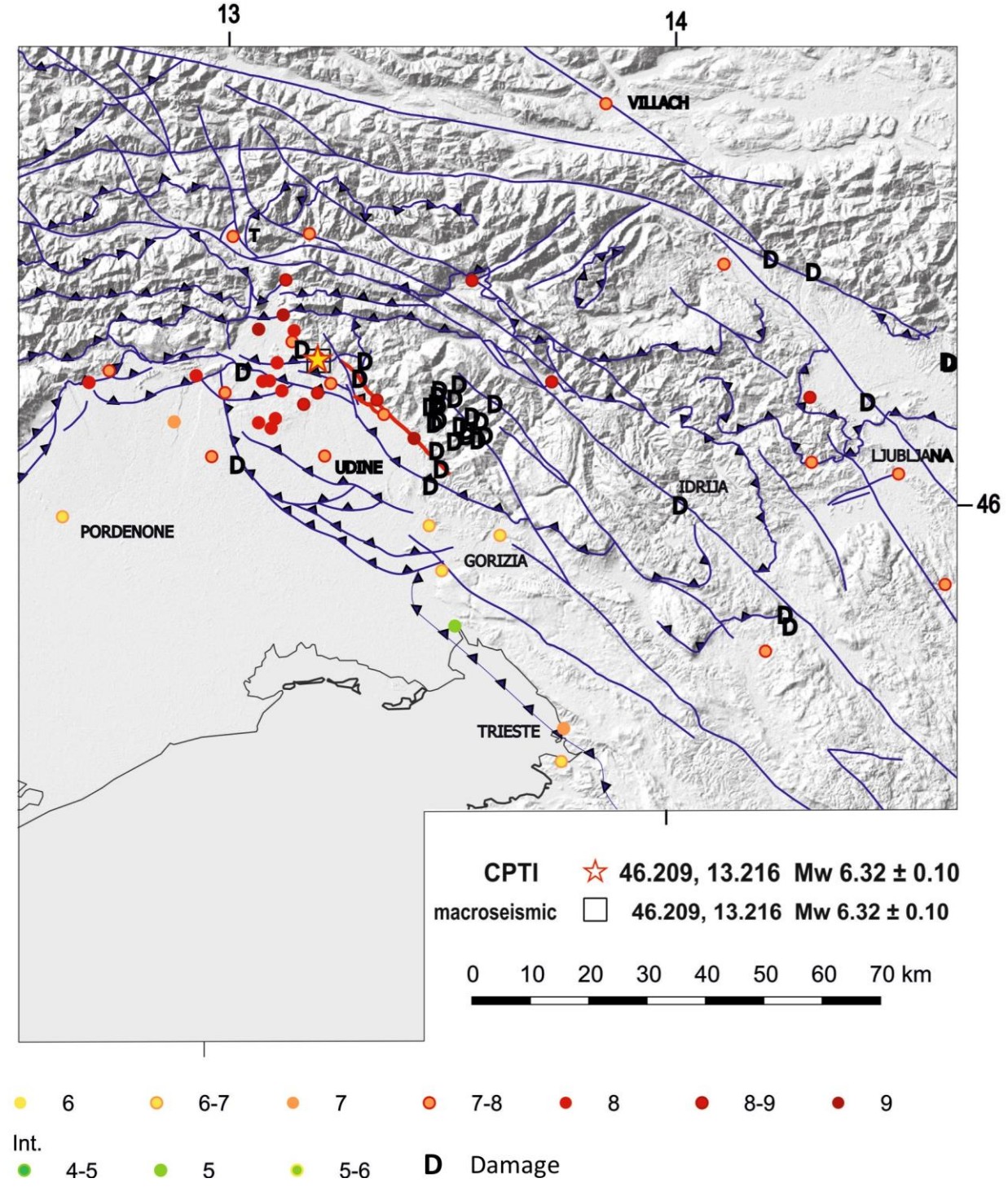

**Figure 8: Damage distribution of the 1511 earthquake from CPTI (Rovida et al., 2016); red lines, BFCF-CVT system.**

| sample code | type of organic material | radiocarbon age (before the present) | calibrated age (2 ɓ) | stratigraphic unit |
|---|---|---|---|---|
| sample_1 | charcoal | 1469±33 | 544-646 AD | unit 4 |
| sample_2 | charcoal | 1503±42 | 526-642 AD | unit 3 |
| sample_3 | charcoal | 6060±79 | 5211-4792 BC | unit 5 |
| sample_4 | charcoal | 291±37 | 1485-1792 AD | unit 2 |
| sample_5 | charcoal | 5596±56 | 4535-4344 BC | unit 5 |
| sample_6 | charcoal | 1225±45 | 674-893 AD | unit 2 |
| sample_7 | charcoal | 1026±75 | 945-1047 AD | unit 2 |

**Table 1: Detail of the radiocarbon dating performed on the collected charcoals (calibration curve by Reimer et al., 2013)**

