# Peer review of "First evidence of active transpressive surface faulting at the front of the eastern Southern Alps, northeastern"

_Solid Earth, 2017_

## Referee Comment (RC1) · PhD Zampieri (Referee) · 21 Feb 2018

Revision of the manuscript

"First evidence of active transpressive surface faulting at the front of the eastern Southern Alps, northeastern Italy. Insight on the 1511 earthquake seismotectonics" by E. Falcucci, M. E. Poli, F. Galadini, G. Scardia, G. Paiero and A. Zanferrari

General comments

The manuscript is a concise description of the results obtained from a paleoseismological trench analysis in the Friuli area (NE Italy), where in 1511 a strong earthquake occurred. This study permits to link the event to a reverse fault, well documented by a seismic line and surficial features. The text is well written, without redundant information. The applied methodology is appropriate, the data and interpretation are convincing, so the work is worth the publication with few improvements and integrations, mainly to the table 2 (see specific comments). This work will impact on the existing knowledge about the 1511 event and the seismic hazard of the close urban area, but it represents also a case study for a wider audience.

Specific comments

Table 2 of the Auxiliary material can be improved by a better organization and can be included in the text. Please draw a true table with columns and rows. Include a column with the laboratory and/or field label of the samples. Insert a column with specification of the type of analysed material (i.e. wood, charcoal, bulk). Please, comment in the text why the ages of the Unit 2 are so different. The ages of the two samples in Fig. 4c are similar (945 AD – 1047 AD and 674 AD – 893 AD), while the age of the sample in Fig. 4a is younger (1485 AD – 1792 AD). The age of the sample from the Unit 3 is very similar to that of the sample from the Unit 4. Could the age of Unit 3 refer to a reworked element?

Technical corrections

p. 1 line 11: a system of NW-SE trending dextral strike-slip faults. line 26: reverse fault ruptures, instead of: inverse ruptures.

p.3 lines 13-14: the BFCF trend is expressed in degrees from north, while the CVT trend is expressed in sectors. Please, uniform. line 20: insert the acronyms for the faults, i.e. PRM and TN. line 20: interpretable, instead of visible. line 30: on the CVT hanging wall.

p. 4 line 1: reverse fault, instead of inverse tectonic structure line 9: clay, not clays line 13: insert a space between Fig. and 2c line 13: place has been found at the end of the sentence. line 27: "Unit 6 is unconformably overlaid". Looking at the figure 4a, in the middle part the boundary between the units 6 and 5 the looks like that of interbedded facies, because it is saw-toothed. Maybe, insertion of some bedding lines can help to unravel the relationships between unit 5 and 6.

p. 4 line 6: a tensional fracture, instead of an extensional fracture.

p.6 line 4: reverse-oblique, instead of inverse-oblique. line 5: shortening instead of compression.

p. 9 line 6 insert "and" in between Gosar, A. Bourlès, D.

Figures

Fig. 1: the label a, b, c are lacking. In b) the fault traces lying in the alluvial plain must be dashed lines (blind faults).

Fig. 2: please, enlarge the inset content on the upper right corner and explain the line drawing symbols (trenches, drill-hole and the arcuate line (is it the trace of the cross section in 2c?)). Specify which are the trenches a, b and c of Fig. 4. Are the three segments on the hanging wall anticline in c) the three trenches? If so, why are they inclined?

Fig. 3: in (a) the black rectangle cited in the caption is lacking. Please, explain also the significance of the curved dashed line.

Fig. 4: The deposit in grey colour infilling the erosional feature incising units 2 and 3 in all trenches is not labelled, nor is it described in the caption. The grey colour in the trenches 1 and 2 is similar, but different from that of trench 3. Are they different deposits?

Fig. 5: the hanging wall fold of the CVT fault in the cross-section is quite different from

the same fold in the 3D scheme. Also the geometry of the faults is different.

Captions

Fig. 1: line 7: lower instead of lowe

Fig. 4: Trench instead of rench

Auxiliary material

Table 1, line 4: "Interpretazione stratigrafiche nel unità poligeniche tagliamento" must be erased

Table 2: Insert a full stop after descrition of Unit 5. Cancel one of the two full stops at the end of description of Unit 7. See also suggestions in Specific comments

---

## Referee Comment (RC2) · L. Wedmore (Referee) · 25 Feb 2018

This paper is concerned with the strain partitioning that is occurring as the Adria microplate converges with the Europe in the eastern Southern (European) Alps. The authors point out that this region has a complex tectonic history which is reflected in the recent seismicity which shows both reverse and strike slip faulting. They dig 3 palaeoseismic trenches across one of the thrust faults in the region, and use 14C radiocarbon dating to constrain the timing of previous earthquakes on the fault. They find 3 separate events in their trenches and correlate the most recent event with an earthquake that has previously been identified by historical shaking records. In doing so, the authors provide convincing evidence that they have determined one of the source faults of the 1511 earthquake.

The authors make a lot of inferences that both structures were active at the same time. The data do not support such a statement as the authors only conducted palaeoseismic analyses on the Colle Villano thrust fault (CVT). The abstract and discussion and concluding remarks should be amended to make this clear. It would be fine to *discuss* the possibility that both the CVT and the Borgo Faris-Cividale fault (BFCF) are active at the same time in accommodating partitioned strain. However, it needs to be clear that the data only support the activity on the CVT.

Although the palaeoseismic trenches across the CVT, and their relationship to the historical earthquakes in the region is the main focus of the I have one question which the authors may wish to address. What evidence is there that the BFCF is still active? There appears to be no clear right-lateral deflection of the rivers that cross the fault. This could be addressed by showing geodetic interseismic strain across the fault if such data already exists (I accept it's beyond the scope of this paper to collect or process such data), or through higher resolution maps of the drainage crossing the BFCF.

In general, the quality of the figures needs improving with often faint lines and difficult to interpret maps.

Line 26: Do the authors mean 'reverse' rather than 'inverse'.

Line 25: You mention the geodetic data here (and comment on GPS time series later in the discussion (page 6, line 5). Whilst you provide the references for this data, it would help the readers to see GPS vectors plotted on a map. These could be added to the top left part of Figure 1 to aid readers in interpreting the tectonics of the area. It would be very useful to see how/if the geodetic strain is partitioned in the same way the authors claim the strain is portioned by the geology (this may also help answer my question about the activity of the BFCF – see above).

Line 15: I am unfamiliar with the term 'mesostructural' please use a simpler term here. Please also describe what sort of kinematic indicators you have plotted in Figure 1.

Lines 7-10: Please give more information about the core you collected. This should include a figure with a detailed core log and photographic examples of the units found in the borehole.

Line 18: extrados is a spelling mistake – this whole sentence doesn't make sense at the moment.
Line 25: capitalise B.C.

Line 23-25. This last sentence is very long and doesn't entirely make sense, particularly the final part of the sentence. Suggested edit:' …where interseismic coupling suggests elastic strain is building up at seismogenic depths which will be released in future large earthquakes.'

Figure 1:
In general I think this figure would benefit from being split into two: the bottom half of the figure (the seismic profile) would fit better in a separate figure where you could show the section both with *and without* the interpretation which would allow the reader to make an informed decision on the validity of their interpretation. Please also place an x-axis on this figure as the current scale makes it difficult to read.
        Top left hand part:  It would be nice to see the focal mechanisms of the recent seismicity actually plotted on the map (rather than in the legend) as this would make the relationship between the strike-slip and thrust faulting clearer. Please include axis on the map indicating the longitude and latitude of the map. In general the lines could be made thicker and it's difficult to differentiate between the different faults and the geographical boundaries. At this scale, a simpler map containing the main tectonic features as well as the recent and historical seismicity would be of benefit to the reader. Consider adding GPS vectors to this map (see earlier comment).

Figure 2:
The inset in part a is difficult to read. This would benefit from being made larger with the location of the palaeoseismic trenches more clearly indicated and the thickness of the contour lines etc increased. Please indicate the source of the digital elevation model.

The axis of part a need improvement: there is a lack of detail and it is not clear what units the map is projected in. Please include a log a details of the bore hole indicated by the yellow dot in part a.

Figure 3:
I know this information is already in the caption to the figure, but it would be helpful if you indicated on the photos themselves which of the trenches is being shown in each photo.

Figure 5
This would benefit from being split into two parts with the conceptual 3D diagrams and the historical earthquake shaking separated. For the historical earthquake shaking figure, please include all major faults in the region as well as the two faults investigated in this paper.

Auxilliary Material:

Both tables could be included in the main text of the paper with little expansion of the length of the article. The formatting of the both tables should be improved. Furthermore, Table 2 requires additional information such as which stratigraphic unit each of the samples has been collected from, the laboratory sample code for each sample, and both the uncorrected 14C age, the calibrated 14C age and the calendar year. Details should be given of the 14C calibration curve used.

---

## Author Comment (AC1) · 2 May 2018

Dear Editors, We have received the revisions that have been suggested for our manuscript "First evidence of active transpressive surface faulting at the front of the eastern Southern Alps, northeastern Italy. Insight on the 1511 earthquake seismotectonics".

In the following pages, please find the details of our comments and the changes we have made to the revised manuscript, along with our answers to the Reviewers to each

point.

We hope that in light of these changes and improvements, you and your Referees will now feel that our manuscript is of sufficient quality and impact for publication in Solid Earth.

We would also like to thank you and your Referees for your comments and suggestions, as we believe that these have permitted us to improve the quality of our research and manuscript.

We look forward to hearing from you further.

Best regards,

Dr. Emanuela Falcucci For and on behalf of all of the Authors

Dear Editor, please find below the answers to all of the Reviewer's comments and suggestions. We have accepted the most, and modified the revised manuscript accordingly. (we list the comments, followed by our answers)

Reviewer 1 (Dario Zampieri):

1) Table 2 of the Auxiliary material can be improved by a better organization and can be included in the text. Please draw a true table with columns and rows. Include a column with the laboratory and/or field label of the samples. Insert a column with specification of the type of analysed material (i.e. wood, charcoal, bulk). Please, comment in the text why the ages of the Unit 2 are so different. The ages of the two samples in Fig. 4c are similar (945 AD – 1047 AD and 674 AD – 893 AD), while the age of the sample in Fig. 4a is younger (1485 AD – 1792 AD). The age of the sample from the Unit 3 is very similar to that of the sample from the Unit 4. Could the age of Unit 3 refer to a reworked element?

Answer: We accepted the Reviewer request. We have now added a table in the main revised manuscript with the details of the achieved radiocarbon dating. We have now

explained in the revised manuscript (paragraph 5) that the obtained ages refer to charcoals, that have been included and transported by the alluvial and colluvial units in which we found them. So, the age can be similar each other or sparse. In light of this, we can only consider the most recent age as a terminus post quem for the unit deposition and, thus, for the deformation events.

2) Technical corrections:

Answer: All of the technical corrections have been accepted. The revised manuscript and the figures have been modified accordingly.

Figure 1 comments: the label a, b, c are lacking. In b) the fault traces lying in the alluvial plain must be dashed lines (blind faults).

Answer: Accepted and now modified.

Figure 2 comments: please, enlarge the inset content on the upper right corner and explain the line drawing symbols (trenches, drill-hole and the arcuate line (is it the trace of the cross section in 2c?) . Specify which are the trenches a, b and c of Fig. 4. Are the three segments on the hanging wall anticline in c) the three trenches? If so, why are they inclined?

Answer: The three black lines in Figure 2c (now figure 4) on the hanging wall of the thrust are not the location of the trenches but they were a simplified representation of the extrados fractures we identified along the trench walls. We have now removed them because hard to understand. Figure 3 comments: in (a) the black rectangle cited in the caption is lacking. Please, explain also the significance of the curved dashed line.

Answer: We have now added a rectangle to indicate the trench siting. We have now also removed the curved dashed line because hard to understand.

Figure 4 comments: The deposit in grey colour infilling the erosional feature incising units 2 and 3 in all trenches is not labelled, nor is it described in the caption. The grey

colour in the trenches 1 and 2 is similar, but different from that of trench 3. Are they different deposits?

Answer: We have now added in the figures the label anthropogenic unit, and modified the colour.

Figure 5 comments: the hanging wall fold of the CVT fault in the cross-section is quite different from the same fold in the 3D scheme. Also the geometry of the faults is different.

Answer: We have now modified the figure to make the cross-section and the 3D scheme comparable.

Captions:

Answer: All of the comments on the figure captions have been accepted and we have now modified them.

[revised manuscript text omitted]

Please also note the supplement to this comment:

https://www.solid-earth-discuss.net/se-2017-131/se-2017-131-AC1-supplement.pdf

[Figure]

[Figure]

**Fig. 1.** Fig. 1

SE                                                                    NW

TWT (ms)

0.0 -

0.5 -

1.0 -

1.5 -

**a**

A                                                                      A'

SE                                                                  NW

TWT (ms)

TN          PRM  CVT          BFCF

0.0 -

Q

UM

FLY

0.5 -

LM                                                     FLY

FLY                    FLY

1.0 -

1.5 -   0    500   1000              3000 m

**b**

A                                                                      A'

3000 m

**Fig. 2.** Fig. 2

[Figure]

**Fig. 3.** Fig. 3

[Figure]

**Fig. 4.** Fig. 4

**Fig. 5.** Fig. 5

**Trench 1 - northern wall**

SW / NE

Anthropogenic excavation and filling

1485-1792 AD [cal., 2sigma]

1, 2, 3, 4, 5, 6, 7, 8

**a**

**Trench 2 - southern wall**

NE / SW

1, 4, 5, 2, 3, 6, 7, 8

Anthropogenic excavation and filling

5211-4792 BC (cal 2 sigma)

526-642 AD (cal 2 sigma)

544-646 AD (cal. 2 sigma)

detail in Fig. 5d

**b**

**Trench 3 - northern wall**

SW / NE

Anthropogenic excavation and filling

1, 2, 3, 4, 5, 6, 7, 8

4535 BC-4344 BC (cal.2 sigma)

945 AD-1047 AD (cal. 2 sigma)

674 AD-893 AD (cal. 2 sigma)

detail in Fig. 5f

**c**

**Fig. 6.** Fig. 6

[Figure]

**Fig. 7.** Fig. 7

[Figure]

**Fig. 8.** Fig. 8

---

## Author Comment (AC2) · 2 May 2018

Dear Editors, We have received the revisions that have been suggested for our manuscript "First evidence of active transpressive surface faulting at the front of the eastern Southern Alps, northeastern Italy. Insight on the 1511 earthquake seismotectonics".

In the following pages, please find the details of our comments and the changes we have made to the revised manuscript, along with our answers to the Reviewers to each

point.

We hope that in light of these changes and improvements, you and your Referees will now feel that our manuscript is of sufficient quality and impact for publication in Solid Earth.

We would also like to thank you and your Referees for your comments and suggestions, as we believe that these have permitted us to improve the quality of our research and manuscript.

We look forward to hearing from you further.

Best regards,

Dr. Emanuela Falcucci For and on behalf of all of the Authors

Dear Editor, please find below the answers to all of the Reviewer's comments and suggestions. We have accepted the most, and modified the revised manuscript accordingly. (we list the comments, followed by our answers)

Reviewer 2 (Luke Wedmore):

1) The authors make a lot of inferences that both structures were active at the same time. The data do not support such a statement as the authors only conducted palaeo-seismic analyses on the Colle Villano thrust fault (CVT). The abstract and discussion and concluding remarks should be amended to make this clear. It would be fine to discuss the possibility that both the CVT and the Borgo Faris-Cividale fault (BFCF) are active at the same time in accommodating partitioned strain. However, it needs to be clear that the data only support the activity on the CVT. Although the palaeo-seismic trenches across the CVT, and their relationship to the historical earthquakes in the region is the main focus of the I have one question which the authors may wish to address. What evidence is there that the BFCF is still active? [. . .] This could be addressed by showing geodetic interseismic strain across the fault if such data already exists (I accept it's beyond the scope of this paper to collect or process such data),

Answer: We understand the Reviewer's concerns about the contemporaneous activity of the Colle Villano Thrust and the Borgo Faris-Cividale Fault and the kinematic relation between them. In the revised manuscript we have now added that pieces of evidence of Late Pleistocene fault activity have been found along the latter structure by other authors, referencing these papers dealing with the morpho-tectonic analyses. This supports that the Colle Villano Thrust and the Borgo Faris-Cividale Fault are active at the same time. As a whole, we have now added in the revised manuscript that, in terms of kinematic relation between the two faults, the following observations suggest that they are the surface expression – as fault splays – of a complex fault system that accommodates transpressive tectonic deformation affecting this region: 1) The narrow spacing (in plan view) between the two structures (no larger than 2 km; towards the south, the two structure merge, as we depicted in figure 1); 2) the deep structural arrangement, achieved by the interpretation of the provided seismic lines, which shows the Colle Villano Thrust as a rather superficial splay that connects to the Borgo Faris-Cividale Fault and does not cut across it; and 3) the evidence of transpressive deformation we observed along the trench walls (testified by both compressive faults and deformations, and sub-vertical strike-slip shear planes), point to the Borgo Faris-Cividale Fault as major strike-slip fault splay, which accommodates the horizontal tectonic deformation, and the Colle Villano Thrust as a synthetic splay that accommodates the contractional component. In order to greet the Reviewer's request, we have now improved the discussion. As for the interseismic deformation from geodetic data, unfortunately, the sparsity of CGPS in the region does not allow robust inferences at the fault scale. Nonetheless, we have now added, as inset in figure 1, the slip vectors defined by D'Agostino et al. (2008), which suggest a transpressive deformation style in the bulk of the region. In the discussion, we have now issued that horizontal shear in the region seems to be also supported by Devoti et al. (2011), figure 3, by the analysis of the regional geodetic strain rate by CGPS time series. In this perspective, it must be also accounted that the contemporary stress map of Italy (Montone and Mariucci, 2016) shows that the area under investigations locates at the transition between

strike-slip faulting and thrust faulting, and transpressive deformation is expected. This consideration has been also now added in the revised manuscript.

2) There appears to be no clear right-lateral deflection of the rivers that cross the fault...

Answer: As for the rivers deflection, we agree with the Reviewer that right-lateral deflection of rivers is faint. It appears from just the already marked streams (see figure 3 in the revised manuscript). As a matter of fact, the other river courses and streams that cross the fault trace in the area are presently man-controlled, artificially-deflected and managed since many decades. The available high-resolution maps or digital terrain models only show the present-day fluvial courses setting, and are therefore not useful to this porpoise. Thus, it is not possible to identify other possible fault-controlled deflections. The right-lateral movements of the Borgo Faris-Cividale fault is indeed issued based on other geologic evidence (Zanferrari et al., 2008; Moulin et al., 2016). We meant fluvial deflections as just a further hint of this.

3) Page 1 Line 26: Do the authors mean 'reverse' rather than 'inverse'

Answer: Accepted and now modified.

4) Page 2 Line 25: You mention the geodetic data here (and comment on GPS time series later in the discussion (page 6, line 5). Whilst you provide the references for this data, it would help the readers to see GPS vectors plotted on a map. These could be added to the top left part of Figure 1 to aid readers in interpreting the tectonics of the area. It would be very useful to see how/if the geodetic strain is partitioned in the same way the authors claim the strain is portioned by the geology (this may also help answer my question about the activity of the BFCF – see above).

Answer: We accept the Reviewer comment and we have now added an inset to figure 1 to show the slip vectors defined by D'Agostino et al. (2008). Unfortunately, as we explained above, the CGPS in the area are rather sparse and no thorough and

reliable inferences at the scale of the fault are possible. Partitioning of slip is manly issued by other authors and by the our work, based on geological evidence. In particular, the evidence we found in the trenches are indicative of tranpressive deformation accommodated by reverse and strike slip shear planes.

5) Page 3 Line 15: I am unfamiliar with the term 'mesostructural' please use a simpler term here. Please also describe what sort of kinematic indicators you have plotted in Figure 1.

Answer: The term was meant to indicate structural data collected at sites where the fault planes or deformation zone are exposed. We have now removed the term. As for the stereo-plot in figure 1b, as we explained in the caption, we plotted striations features collected along the fault shear zone.

6) Lines 7-10: Please give more information about the core you collected. This should include a figure with a detailed core log and photographic examples of the units found in the borehole.

Answer: We accept the Reviewer's comment. We have now added a new figure with the core log and some pictures of the units found in the borehole.

7) Page 5 Line 18: extrados is a spelling mistake – this whole sentence doesn't make sense at the moment.

Answer: Extrados is a term that is commonly used to indicate tensional fractures connected to bending moment faulting. We have now added this esplanation.

8) Page 6 Line 23-25: This last sentence is very long and doesn't entirely make sense, particularly the final part of the sentence. Suggested edit:' …where interseismic coupling suggests elastic strain is building up at seismogenic depths which will be released in future large earthquakes.'

Answer: We accepted the suggested edit.

Figures:

Figure 1: In general I think this figure would benefit from being split into two: the bottom half of the figure (the seismic profile) would fit better in a separate figure where you could show the section both with and without the interpretation which would allow the reader to make an informed decision on the validity of their interpretation. Please also place an x-axis on this figure as the current scale makes it difficult to read. Top left hand part: It would be nice to see the focal mechanisms of the recent seismicity actually plotted on the map (rather than in the legend) as this would make the relationship between the strike-slip and thrust faulting clearer. Please include axis on the map indicating the longitude and latitude of the map. In general the lines could be made thicker and it's difficult to differentiate between the different faults and the geographical boundaries. At this scale, a simpler map containing the main tectonic features as well as the recent and historical seismicity would be of benefit to the reader. Consider adding GPS vectors to this map (see earlier comment).

Answer: We accepted all of the Reviewer's suggestions. We have now also added an inset to the figure with the slip vector derived from GPS time series.

Figure 2: The inset in part a is difficult to read. This would benefit from being made larger with the location of the palaeoseismic trenches more clearly indicated and the thickness of the contour lines etc increased. Please indicate the source of the digital elevation model. The axis of part a need improvement: there is a lack of detail and it is not clear what units the map is projected in. Please include a log a details of the bore hole indicated by the yellow dot in part a.

Answer: We have now improved the readability of the figures and added what was lacking. We have now also added a new figure (figure 4) with the details of the bore hole.

Figure 3: I know this information is already in the caption to the figure, but it would be helpful if you indicated on the photos themselves which of the trenches is being shown

in each photo.

Answer: We accepted the Reviewer's suggestion. We have now modified the figure accordingly.

Figure 5 This would benefit from being split into two parts with the conceptual 3D diagrams and the historical earthquake shaking separated. For the historical earthquake shaking figure, please include all major faults in the region as well as the two faults investigated in this paper.

Answer: We accepted the Reviewer's comment. We have now splitted the figure into two new figures, including all major faults and those investigated in our work in the new figure.

Auxilliary Material: Both tables could be included in the main text of the paper with little expansion of the length of the article. The formatting of the both tables should be improved. Furthermore, Table 2 requires additional information such as which stratigraphic unit each of the samples has been collected from, the laboratory sample code for each sample, and both the uncorrected 14C age, the calibrated 14C age and the calendar year. Details should be given of the 14C calibration curve used.

Answer: We have now moved the units description in the main text and we have now improved and added to the main text the new Table 1, with the details of the radiometric age determinations. Details of the calibration curve has been added in the figure caption.

[revised manuscript text omitted]

Please also note the supplement to this comment:
https://www.solid-earth-discuss.net/se-2017-131/se-2017-131-AC2-supplement.pdf

———————————————————

[Figure]

[Figure]

**Fig. 1.** Fig. 1

The seismic section figure showing profile A–A' from SE to NW with interpreted geological units (Q, UM, LM, FLY) and fault labels (TN, PRM, CVT, BFCF).

**Fig. 2.** Fig. 2

[Figure]

**Fig. 3.** Fig. 3

**Fig. 4.** Fig. 4

![Figure 5 composite of photographs a–f]

**Fig. 5.** Fig. 5

**Trench 1 - northern wall**

SW ... NE

1485-1792 AD [cal., 2sigma]

Anthropogenic excavation and filling

a

**Trench 2 - southern wall**

NE ... SW

Anthropogenic excavation and filling

5211-4792 BC (cal 2 sigma)

526-642 AD (cal 2 sigma)

544-646 AD (cal. 2 sigma)

detail in Fig. 5d

b

**Trench 3 - northern wall**

SW ... NE

Anthropogenic excavation and filling

945 AD-1047 AD (cal. 2 sigma)

674 AD-893 AD (cal. 2 sigma)

4535 BC-4344 BC (cal.2 sigma)

detail in Fig. 5f

c

**Fig. 6.** Fig. 6

**Fig. 7.** Fig. 7

[Figure]

**Fig. 8.** Fig. 8